# Intergenerational Ties in Context: Association between Caring for Grandchildren and Cognitive Function in Middle-Aged and Older Chinese

**DOI:** 10.3390/ijerph18010021

**Published:** 2020-12-22

**Authors:** Shiming Liao, Ling Qi, Jie Xiong, Jie Yan, Ruoxi Wang

**Affiliations:** 1School of Public Health, Fudan University, Shanghai 200032, China; 20211020204@fudan.edu.cn; 2School of Health Science and Nursing, Wuhan Polytechnic University, Wuhan 430023, China; qiling_2017@whpu.edu.cn; 3ESSCA School of Management, 1 Rue Joseph Lakanal-BP 40348, CEDEX 01, 49003 Angers, France; jie.xiong@essca.fr; 4Grenoble Ecole de Management, 12 Rue Pierre Semard, 38000 Grenoble, France; jie.yan@grenoble-em.com; 5School of Medicine and Health Management, Tongji Medical College, Huazhong University of Science and Technology, Wuhan 430030, China; 6Research Center for Rural Health Services, Hubei Province Key Research Institute of Humanities and Social Sciences, Wuhan 430030, China

**Keywords:** grandparent caregiving, cognitive function, gender differences, China

## Abstract

Grandchild caregiving is suggested to improve the elderly’s cognitive function, but the specific relationship remains under-investigated. Considering gender disparity, this study aimed to understand the relationship between grandchild caregiving and cognition. In total, 7236 Chinese residents (≥45 years old) were selected from the 2015 China Health and Retirement Longitudinal Study (CHARLS). The China Health and Retirement Longitudinal Study Harmonized Cognitive Assessment Protocol (CHARLS-HCAP) was used to measure cognition. Grandparenting was measured from three dimensions: caregiving frequency, intensity, and the number of grandchildren cared for. The relationship was examined by multivariate linear regression, with age as a moderator. The results showed that the majority of respondents provided care to their grandchildren, especially grandmothers. Grandchild caregiving was positively associated with cognition (β = 0.686, 95% CI = 0.334–1.038), especially in the older-aged group. Moderate, not regular grandparenting, or caring for one grandchild was more positively associated with cognitive function. However, intensive and regular grandchild care was significantly associated with cognition only in men. No moderating effects of age were found in women. The study confirmed that moderate intensity and frequency of caregiving was related to better cognitive function in middle-aged and older Chinese population, whereas cultural context and gender differences could be considered when designing targeted policies.

## 1. Introduction

With the global population aging [1], the age-related cognitive decline [2,3] has become a worldwide public health issue. It not only damages physical and mental health, increasing the economic, physical, and psychological burden of family caregivers [4,5], but also increases the cost to family, community, and government [6]. Moreover, dementia is diagnosed when cognitive impairment develops seriously enough to affect daily activities [7,8], and most of the dementia will occur in low- and middle-income countries (LMICs) [9]. In China, as one of the LMICs [10] and the most populous country in the world [11], cognitive impairment has become a severe health problem for the elderly population. However, like other LMICs, China also lacks sufficient professional mental health resources [12,13] due to the high demand for professional resources [14,15], along with a relatively low public awareness rate of cognitive impairment [16]. Therefore, China urgently needs a more widespread and cost-effective way to address this issue.

Caring for grandchildren has both broad applicability and potential effectiveness. Numerous studies have suggested that social connections and productive activities are beneficial not only to physical health, e.g., helping treat chronic diseases [17] but also to mental health, e.g., improving cognitive function in the elderly [18,19,20]. As one of the productive activities [21,22], grandparent caregiving is more common for the Chinese elderly due to China’s special social [23] and cultural [24] background. Moreover, considering the fact that grandparent caregiving has a gender-specific nature that women are seen as primary caregivers [25], it is customary for grandmothers to provide care to grandchildren, especially in intensive grandchild care [26].

Some studies have proposed that grandparents may bear different role-taking mechanisms during their caregiving activities, namely “role enhancement theory” [27] or “role strain theory” [28]. The relationship between grandparent caregiving and cognition in the middle-aged and older adults has not been determined. Studies have shown that caring for grandchildren may maintain the cognition of the elderly due to its positive nature [29]. Grandparent caregiving may be a huge demand on caregivers, both socially and emotionally, and may require relatively large cognitive resources [30], which could delay cognitive function decline [31]. Similarly, caring for grandchildren could increase grandparents’ daily activities [32], such as having fun with grandchildren to encourage physical activities and help them feel more energetic and young, thereby increasing their social participation and improving cognition [33,34].

Nevertheless, a negative association between grandparent caregiving and cognition was also observed. Some studies found that grandparent caregiving may cause stress, limit other forms of social participation, and reduce self-care in the elderly, thus negatively affecting cognitive function and mental health [35,36]. Moreover, a handful of research further investigated the frequency [36], intensity [37], number of grandchildren [38], and cohabitation of grandparent caregiving [39], but also failed to reach an agreement on the optimal patterns of grandparenting that can contribute to cognition [40,41,42]. In other words, caring for grandchildren and its relationship with cognition is a multidimensional concept that needs further study. Meanwhile, most research is conducted in Western countries, leaving China and other LMICs uninvestigated.

Given this, we hypothesized that a positive association between grandparent caregiving and cognitive function among the middle-aged and elderly Chinese existed. Thus, our study aimed to investigate the patterns of grandparent caregiving and explore its relationship with cognitive function from three dimensions, including intensity, frequency, and the number of grandchildren they cared for. Furthermore, considering the gendered nature of grandparent caregiving that compared with men, women play a more important role in the care of grandchildren [43,44,45], and the potential interactions between age and grandparenting, this study was carried out with gender differences and the moderating effect of age taken into consideration.

## 2. Materials and Methods

### 2.1. Study Sample

The data were extracted from the China Health and Retirement Longitudinal Study (CHARLS) Wave 4 (2015) survey [46], which is a nationally representative longitudinal survey. The CHALRS collects the demographic and socioeconomic data of Chinese residents aged 45 years and above, as well as health behaviors and health-related outcomes. The baseline survey was conducted in 2011 and followed up in 2013, 2014, and 2015. Using a multistage probability-proportional-to-size (PPS) sampling technique [46,47], CHARLS surveyed with 28 provinces, 450 villages or communities in 150 districts or counties covered, and 21,097 individuals aged 45 years and older in 12,221 households were investigated in 2015 [48]. Considering that grandparent caregiving may have contemporaneous impacts on mental health and some short-term effects may diminish over time [49], we only employed Wave 4 data and selected 7236 participants according to the following criteria: (1) aged 45 and older; (2) provided information on both grandparent caregiving and cognitive function.

### 2.2. Variables

#### 2.2.1. Outcome Variables

Cognitive function was measured by the China Health and Retirement Longitudinal Study Harmonized Cognitive Assessment Protocol (CHARLS-HCAP) [50], which was similar to the U.S. Health and Retirement Study (HRS) [51] and has been proven feasible and valid to be used in the CHARLS sample and hospital samples [50]. Similar to the HRS, the CHARLS-HCAP combined items from several scales, including Mini-mental State Examination (MMSE) and Consortium to Establish a Registry for Alzheimer’s Disease (CERAD). The CHALRS adopts Date Naming, Serial 7′s test, and drawing of overlapping pentagons from MMSE [50], and adopts the CERAD version’s key components of immediate and delayed word recall to measure memory [52]. As a result, considering that the measures in CHARLS included components of different scales and referring to previous studies using the CHARLS data [53,54,55], no cutoffs have been employed in our study.

Based upon previous studies using the data from HRS [51] and CHARLS [53,54,55], respectively, we assessed the two dimensions of cognitive function separately: episodic memory and mental status (Table 1). Specifically speaking, the tests in CHARLS measured four aspects of cognitive function, ranging from 0 to 31 points, representing the overall cognitive status of the respondents: episodic memory, orientation, visuoconstruction, and mathematical performance [56]. The higher the scores, the better the cognitive function [55]. In CHARLS, episodic memory was assessed by immediate and delayed word recall, which was to test respondents’ capacity to repeat after reading ten Chinese nouns in any order immediately and four minutes later, respectively. Both of them ranged from 0 to 10 points. By summing the immediate and delayed recall scores, the total score of episodic memory was 20 points. Mental status consisted of orientation, visuoconstruction, and mathematical performance, ranging from 0 to 11 points. Orientation was measured using a five-item scale that asks participants to name the day, month, year, season, and correct day of the week. Visuoconstruction was measured by asking participants to re-draw a previously shown picture accurately (0–1 point). The mathematical performance was assessed by asking participants to subtract 7 from 100 up to 5 times (0–5 points).

#### 2.2.2. Explanatory Variables

In the CHARLS 2015 questionnaire, grandparent caregiving was measured by the following questions: (1) Did you spend time caring for your grandchildren last year; (2) if yes, which child’s children did you provide care for; (3) how many weeks and how many hours per week did you spend caring for this child’s children approximately.

Intensity: according to the respondents’ answers on how many hours they spent on caring for grandchildren per week, the average number of hours the grandparents looked after their grandchildren was 33.4 h per week, roughly equivalent to holding a full-time job. By referring to the previous study that used a standard working time of 5 days a week and 8 h a day as a cutoff [57], and based on the principle of averaging the sample size of each category, this continuous variable was divided into three categories: Never/Moderate grandchild care (1–39 h)/ Intensive grandchild care (≥40 h).

Frequency: according to the respondents’ answer about how many weeks a year they spent caring for their grandchildren, the average number of weeks the grandparents cared for their grandchildren was 19 weeks a year, roughly equivalent to half a year. Considering that some participants had no caregiving activities, this continuous variable was divided into three categories according to the principle of averaging the sample size of each category: Never/Not regularly (≤half a year: 1–26 weeks)/Regularly (≥half a year: ≥26 weeks).

Number of grandchildren: according to the respondents’ answers on which grandchildren they took care of and considering that some participants had no caregiving activities, the continuous variable was divided into three categories based upon the principle of averaging the sample size of each category: 0/1/≥2.

#### 2.2.3. Control Variables

Table 2 presents the codes and (or) definitions of all control variables referred from prior studies [58,59,60].

### 2.3. Data Analysis

Percentages and frequencies were calculated for descriptive data. Chi-square test for categorical variables and Wilcoxon rank sum test and independent sample T-test for continuous variables were performed to test gender differences. One-way ANOVA test was performed to analyze the cognition performance in different grandchild caregiving patterns. After controlling for potentially confounding variables, multivariate linear regression analysis was employed to assess the association between grandparent caregiving and cognition in the whole sample and each gender group. In order to avoid the potential multicollinearity, different caregiver types were entered separately as a single explanatory variable. Additionally, whether relations were moderated by age was examined. A moderating effect was indicated by a significant interaction of grandchildren caregiving patterns × moderator age. In all regression analyses, the predictor variables were mean-centered. Coefficients (β1) and their 95% confidence intervals (95% CIs) were employed to depict the associated effect. The data were analyzed by R Version 3.5.1 (Bell Laboratories, Wien, Austria).

## 3. Results

### 3.1. Basic Characteristics of the Participants

Table 3 shows the basic characteristics of the whole participants, as well as male/female subgroups. The median age was 62 years old. Of the 7236 participants, greater proportions were rural residents (62.10%), with a primary school or lower level of education (44.03%), having a partner (79.91%), and not retired (65.10%). Moreover, most participants did not drink alcohol (63.90%) or smoke (70.10%); suffered from at least one type of NCD (82.90%), experienced no difficulty in activities of daily living (78.50%); lived near children (54.20%), had weekly contact with children (78.30%), and received economic support (93.00%) from them.

As shown in Table 3, gender differences in sociodemographic characteristics and health status were also significant. Compared with women, men had a higher education level, a greater proportion of alcohol and tobacco consumption, but a lower retirement rate, and suffered less from chronic disease and activities of daily living (ADLs). Furthermore, men scored significantly higher than women in cognition (15 vs. 13). Moreover, a lower proportion of men lived with their children, received economic support, and contacted with them than women.

### 3.2. Patterns of Grandparent Caregiving

Table 4 presents the frequency, intensity, and the number of grandchildren respondents cared for. In general, over half of respondents cared for their grandchildren (52.90%). Amongst those who provided care, 26.19% provided care not regularly, with the intensity of 40 h and above a week (29.53%) to one grandchild (42.03%). Significant differences between men and women were also observed. Compared with male counterparts, women were the primary caregivers for their grandchildren regarding frequency (24.65% vs. 21.30% care regularly) and intensity (32.88% vs. 26.46% intensive grandchild care).

### 3.3. Cognitive Function in Different Patterns of Grandparent Caregiving

Table 5 presents the mean and standard deviation (SD) of cognitive function in different grandparent caregiving patterns, as well as the cognitive differences in specific caregiving patterns. In general, significant differences in cognitive function existed among different patterns of grandchildren caregiving. The respondents who cared for their grandchildren had higher cognitive scores than their counterparts with no caregiving activities, and the trend was the same in the two gender subgroups.

### 3.4. Relationship between Grandparent Caregiving and Cognitive Function

Results from the multivariate linear regressions of the relationship between grandparent caregiving and cognition for all participants and each gender group are reported in Table 6. Broadly speaking, caring for grandchildren was significantly associated with cognition. Providing moderate or not regular grandchild care was more strongly associated with cognition than intensive or regular grandchild care. The trend was similar to the trend for the number of grandchildren cared for, in that, compared with taking care of two or more grandchildren, caring for only one grandchild was more positively associated with cognitive function.

Similar relationships were found between grandchild care and cognitive function in different gender subgroups. A positive association between grandparent caregiving and cognition was significant in both men and women, with lower intensity, lower frequency, and one grandchild to care for. However, providing intensive and regular grandchild care was significantly associated with cognition only for male respondents.

Interaction analysis revealed that in the whole sample, age moderated the positive associations between grandchildren care and cognition (Table 6). These associations were more pronounced in the older aged group, especially among those who provided moderate care to one grandchild. Similar relationships were also found in the male subgroup. However, we did not find such a moderating effect in the female subgroup.

Furthermore, by referring to previous studies [62,63,64], we employed R^2^_adjusted_ and variance inflation factor (VIF) to measure the effect size and multicollinearity of the multivariable linear regression model, respectively. In all three models, the value of R^2^_adjusted_ > 0.26 and VIF < 10, which suggested that the effect size was relatively large [65] and the strength of association between dependent and predictable variables was moderate [63], as well as that the multicollinearity was weak in our three models [64].

## 4. Discussion

This study advances the knowledge on the relationship between grandparent caregiving and cognition function in the middle-aged and older Chinese population, based on which, we further conducted a detailed analysis of the association between grandparent caregiving and cognition within each gender group, as well as considering three dimensions of care: frequency, intensity, and the number of grandchildren.

### 4.1. Prevalence of Grandparent Caregiving

The finding outlined a high prevalence of grandparent caregiving in the middle-aged and older Chinese population (≥50%), which is similar to the results from research conducted in China [42] and other countries [66,67]. The underlying reason might be that, with more mothers working in the labor market and rising separation and divorce levels, grandparents are now playing an increasingly significant role in child caregiving [68,69]. Additionally, rooted in the filial piety of Confucian norm [70], it is more common for older adults to live with their children and grandchildren in China [71], whereas the majority of grandparents in Western countries conform to a norm of non-interference in intergenerational relationships and do not play a central role in caring their grandchildren [72]. Moreover, although dramatically changing as a result of China’s socioeconomic development, the historically strong tie between grandparents and grandchildren has persisted [73,74]; as more young adults migrate to search for better economic opportunities, a large number of children have been left behind and are often cared for by their grandparents [75]. This suggests that caring for grandchildren is common and plays an active role in China’s childcare system.

### 4.2. The Relationship between Grandparent Caregiving and Cognition

The present study found a significantly positive association between grandparent caregiving and cognitive function. Similar findings can be retrieved from previous studies [37,76]. We explain our findings based on three potential cognitive mechanisms: first, enhance cognition by improving physiological functionality [77]. To be more specific, the underlying reasons might be as follows: First, it increases mental stimulation and therefore enhances brain function [78,79]. During the process of caretaking activities, such as living arrangements for grandchildren and the direct or indirect transmission of individual experience and knowledge, they might strengthen their mental stimulation and cognitive function by practicing their cognitive skills in learning, thinking, and reasoning [80]. Second, grandparents perform various grandparental roles—prior studies have categorized them into four major functions: fun seeker, daily life helper, the reservoir of family wisdom, and surrogate parent [81,82,83]. In this case, grandparent caregiving might have different impacts on cognitive function through various caretaking activities. For instance, through daily life activities such as cleaning the house, washing clothes, feeding, dressing, and bathing, grandparents’ daily activity levels could be improved [84], as well as their mobility and physiological functionality [85]. Third, it mitigates the stress-related impact on cognition [2]. For older adults, grandparenthood can be a meaningful source of leisure [86], which provides them with enjoyable experiences and chances to develop relationships with family members, gives a sense of purpose, and is a source of personal satisfaction or growth [87,88], and even reduces the risk of depression [89]. This could also be supported by the role enhancement theory that grandparents who help to care for their grandchildren could promote an active lifestyle and gain emotional and social support by interacting with others [90], as well as attain social gratification and integration from their various social roles [91]. The demand for multiple roles could help increase social support and offset the risks of role strain [92], which might mitigate the stress-related impact, and as a result, be helpful to maintain cognitive function status.

On the other hand, some previous studies have suggested that dementia or other cognitive impairment could cause disability and affect individuals’ capacity to accomplish daily routines. This may result in the inability to live independently [78,93] and provide care for their grandchildren. Some studies also suggested that the prevalence of depression in older adults with cognitive impairment was higher than that in the general population, regardless of age, race, or gender [94,95]. Consequently, the opposite direction of causality could also be considered, and there is a need for further research.

We also investigated the relationship between patterns of grandparent caregiving and cognition. The results showed that, although intensive and regular grandchild care was positively associated with cognition, providing lower-intensive and frequent grandchild care to fewer grandchildren were more strongly related to cognitive function, which is similar to the findings in prior studies [36,38,96,97]. The reasons might be that intensive and regular caregiving requires grandparents to invest more time and energy, which could cause physical burden and take away resources from them, and consequently, brings more pressure and responsibilities to grandparent caregivers [32,35], compromises physical and mental health [98], and to some extent, offsets the good effects characterized by the grandparental role [99].

### 4.3. Gender Differences

To sum up, significant gender differences existed in cognitive status, grandparent caregiving patterns, and its association with cognition.

Significant differences in cognitive function status in men/women subgroups were observed. Compared with female respondents, male respondents scored significantly higher, which can be supported by prior research conducted in China [53,100]. Inspired by previous studies that a positive correlation between socioeconomic status (SES) and mental health existed [101], this may be because levels of education [102] and income [103] are positively related to cognitive function, and in China’s context, Chinese females have historically lower levels of education and income [53], and retire from jobs earlier than men [45], which could result in female disparity in cognition. This finding reveals that women’s cognitive function is relatively poor and more urgent may be needed help to enhance female cognitive function.

In addition, we also revealed different patterns of differences by gender that grandmothers provided more intensive and regular care to grandchildren than grandfathers, which is consistent with a previous study [104]. This could be explained partly by the grandmother hypothesis [44,105] that in the case of older females whose fertility declines, especially post-menopausal women who are nearly infertile, they tend to invest resources to raise their grandchildren rather than continue to reproduce themselves. This care could have not only endowed their daughters with greater fertility to resume reproduction sooner but also increased their own fitness and benefited their more prolonged survival [106]. Moreover, in China, women often have lower education and income levels, retire earlier than men, and often bear the major responsibility for nurturing family members by social norms [45]. As a result, women are in relatively lower socioeconomic groups and are less able to say no to grandparent caregiving [107]. Therefore, they may invest more time and effort to look after their grandchildren. This echoes our proposition that gender differences should be considered when investigating the patterns of grandparent caregiving and their relationship with cognition in the middle-aged and older Chinese population.

Our study further examined the correlation between patterns of grandparent caregiving and cognitive function in middle-aged and older Chinese residents, and no significant differences were observed in gender groups. Caring for grandchildren was positively statistically related to cognition in men and women, mainly manifested in moderate and irregular care for one grandchild. The reason might be the same with the discussion in the whole sample. Interestingly, providing intensive and regular care to grandchildren was significantly related to cognition only in male respondents. We speculate that, on the one hand, this may be due to different activities involved in grandparent caregiving. Regarding intensive grandparent caregiving, grandfathers usually play the role of a playmate, fun-seeker, and companion, while grandmothers more possibly undertake more intensive responsibilities, such as feeding, dressing, and bathing [108]. Hence, grandmothers may feel more stressed and powerless in the face of intensive caregiving. On the other hand, differences in personality characteristics between men and women are also significant, in that, women are emotion-oriented or interpersonally oriented regarding coping strategies, revolving around expressiveness and sensitivity [109], while men are usually task-oriented [110]. Grandmothers generally contact more frequently and report greater closeness with their grandchildren than grandfathers [111]. They tend to care for grandchildren more directly and spend more time with them, which may bring them more burden and pressure from grandparent caregiving. Moreover, compared with men, women are more susceptible to stress than men, and their stress response is stronger [112,113]. Meanwhile, a physiological stress response could be caused by stress, which is thought to be the biological basis for the link between stress and cognitive function. The hypothalamic–pituitary–adrenal axis could be activated, and the stress hormone cortisol is secreted in the stress response. Cortisol can cross the blood–brain barrier into the brain and bind to receptors in different brain areas, such as the frontal lobe, hippocampus, and amygdala, which are all related to cognitive functions, such as learning and memory [114,115,116]. Additionally, stress is closely related to depression [117], and depression is one of the risk factors of cognitive decline [118]. To some extent, intensive care for grandchildren may counteract the beneficial effect on women’s cognitive function, and intensive care of grandchildren was only related to cognition in male grandparents.

The present study also revealed that women had worse cognitive function and were more likely to provide intensive grandchild care. However, during the process of intensive grandchild care, this cognitive vulnerability may be underscored by grandmothers’ tendency to ignore their own health in favor of their grandchildren [119], and to some extent, counteracts the positive effects of caring for grandchildren on cognitive function. The study, therefore, reminds policymakers to consider gender differences when making policies regarding improving and maintaining cognitive function.

### 4.4. Moderating Effect of Age

In general, the association between grandparenting and cognitive performance was closer in older adults than their middle-aged counterparts. This was especially the case for those who cared for fewer grandchildren with lower intensity. We further explored gender differences and found out that the moderating effects of age held in men, whereas no moderating role of age was observed in women. The possible reason might be that their social interaction became relatively less and limited as they aged, especially after retirement. As one of the very few types of social interactions, grandchild caregiving becomes a more important activity that could help improve their mobility and increase positive health behaviors [119], as well as expand social participation and develop an active lifestyle [120]. This may have positive effects on the health and cognitive function of older adults [41,42]. Compared with male counterparts, women may have more opportunities to take part in different types of social activities after retirement, such as square dance [121] or mahjong [122], and taking care of their grandchildren is one of the ways to socialize, thus it may counteract the positive effects of grandparenting on cognition.

### 4.5. Strength and Limitations

To the best of our knowledge, this study is the first one to investigate the relationship between caregiving for grandchildren and cognition in middle-aged and older Chinese people from various dimensions, including intensity, frequency, and the number of grandchildren cared for. Meanwhile, it is also one of the very few studies that examined gender disparities, which is particularly important considering the gendered nature of cognition and grandparenting patterns. Our findings add further insights into the correlation between the specific patterns of grandparent caregiving and older adults’ cognitive function in each gender group. They not only inform that lower intensity, frequency of care, and fewer number of grandchildren cared for may be more beneficial to cognition, but also remind that social programs and interventions for preventing cognitive aging should be designed with gender differences taken into consideration.

The study also has the following limitations. First, causal relationships could not be inferred, although considering the contemporaneous impacts of grandparent caregiving on mental health [49]. Second, grandparent caregiving variables were self-reported and measured in the last year, which may have some recall bias. Third, whether grandparents co-habited with their grandchildren, what kind of activities grandparents participated in, or what type of role the grandparents fulfilled was not studied due to the data availability, which may also affect cognitive function. Fourth, due to the limited number of older participants caring for their grandchildren, the relationship between grandparent caregiving and cognitive function in the older-aged population in China might not have been fully represented. Lastly, the CHARLS-HCAP is a self-report screening tool rather than a clinical diagnostic measure. The extent of cognitive impairment or the clinical diagnosis of the cognitive impairment could be not obtained.

## 5. Conclusions

This study has yielded three main findings: (1) Caregiving for grandchildren was related to better cognitive function, whereas the magnitude depended largely on the frequency and intensity of care. (2) Cognitive function and patterns of caregiving varied by gender subgroups, where women were primary grandchildren caregivers and suffered from worse cognitive function than men. The positive association between caregiving and cognitive function generally held similar in two subgroups, except that a statistically significant association between intensive and regular grandchildren care and cognitive function was only found in men. (3) A moderating effect of age on the association between grandchild caregiving and cognition was found, especially in the older-aged group who provided moderate care to one grandchild. Similar relationships were found in men, whereas no such moderating effect was observed in women.

## Figures and Tables

**Table 1 ijerph-18-00021-t001:** Items adopted for the neuropsychological tests for China Health and Retirement Longitudinal Study Harmonized Cognitive Assessment Protocol (CHARLS-HCAP).

Cognitive Function	Items	Total Scores
Episodic Memory	Immediate word recall: by asking participants to repeat after reading ten Chinese nouns in any order immediately	10
	Delayed word recall: by asking participants to repeat after reading ten Chinese nouns in any order four minutes later	10
Mental Status	Orientation: by asking participants to name the day, month, year, season, and correct day of the week	5
	Visuoconstruction: by asking participants to re-draw a previously shown picture accurately	1
	Mathematical performance: by asking participants to subtract 7 from 100 up to 5 times	5
Total		31

**Table 2 ijerph-18-00021-t002:** Codes/definition of the control variables.

Variable	Codes/Definition
Gender	1 = Male; 2 = Female
Age	≥45 years, calculated by the respondent’s birth year and month minus the interview year and month
Residence *	0 = Urban; 1 = Rural/Residence is defined by household living region. Whether the region was rural or urban was defined by the National Bureau of Statistics of the People’s Republic of China
Education	1 = illiterate; 2 = ≤primary school; 3 = middle school; 4 = ≥high school
Marital Status	1 = divorced/widowed/single; 2 = married/cohabitating
Retired	0 = No; 1 = Yes
Personal Income	Yuan, the sum of all personal income of respondents
Drinking	0 = No; 1 = Yes, if the respondent has had an alcoholic beverage in the last 12 months
Smoking	0 = No; 1 = Yes, if the respondent is still smoking currently
Number of Types of Chronic Diseases (NCDs)	0 = none; 1 = one type of chronic disease; 2 = two types of chronic disease; 3 = three types of chronic disease and above/chronic diseases included whether the respondent reported having hypertension, diabetes or high blood sugar, cancer, chronic lung disease, heart problem, stroke, psychiatric problem, and arthritis
Activities of Daily Living (ADLs)	0 = No; 1 = Yes, if the respondent experiences any difficulty dressing, bathing and showering, eating, getting in and out of bed, using the toilet, and controlling urination and defecation
Living Near Children	0 = No; 1 = Yes, if any child is co-residing with the respondent and his/her spouse or partner
Economic Support	0 = No; 1 = Yes, if the respondent or spouse has received economic support from any of their children in the past year
Weekly Contact	0 = No; 1 = Yes, if a respondent or spouse has any weekly contact with any of their children in person

* The definition of residence was quoted from the Harmonized CHARLS Documentation [61].

**Table 3 ijerph-18-00021-t003:** Sample characteristics of the selected participants.

Variable	Total	Men	Women	*p*
	(*n* = 7236)	(*n* = 3879)	(*n* = 3357)	
	*n* (%)	*n* (%)	*n* (%)	
Age				<0.001 ^b^
Median (Min, Max)	62 (45, 101)	63 (45, 92)	60 (45, 101)	
Residence				<0.001 ^a^
Urban	2743 (37.9)	1376 (35.5)	1367 (40.7)	
Rural	4493 (62.1)	2503 (64.5)	1990 (59.3)	
Education				<0.001 ^a^
Illiterate	1107 (21.8)	275 (10.1)	832 (35.5)	
≤Primary school	2237 (44.0)	1297 (47.4)	940 (40.1)	
Middle school	1117 (22.0)	721 (26.4)	396 (16.9)	
≥High school	620 (11.2)	442 (16.2)	178 (7.6)	
Missing	2155	1144	1011	
Marital Status				<0.001 ^a^
Divorced/widowed/single	1454 (20.1)	479 (12.3)	975 (29.0)	
Married/cohabitating	5782 (79.9)	3400 (87.7)	2382 (71.0)	
Retired				<0.001 ^a^
No	4622 (65.1)	2694 (70.7)	1928 (58.6)	
Yes	2476 (34.9)	1114 (29.3)	1362 (41.4)	
Missing	138	71	67	
Personal Income				0.155 ^b^
Median (Min, Max)	2640 (−65,775, 650,150)	2605 (−65,775, 302,562)	2700 (−8450, 650,150)	
Missing	3611	1917	1694	
Drinking				<0.001 ^a^
No	4624 (63.9)	1766 (45.6)	2858 (85.2)	
Yes	2608 (36.1)	2111 (54.4)	497 (14.8)	
Missing	4	2	2	
Smoking				<0.001 ^a^
No	5072 (70.1)	1904 (49.1)	3168 (94.4)	
Yes	2159 (29.9)	1971 (50.9)	188 (5.6)	
Missing	5	4	1	
Number of Types of NCDs				<0.001 ^a^
0	1015 (17.1)	608 (18.9)	407 (14.8)	
1	1449 (24.3)	820 (25.5)	629 (22.9)	
2	1324 (22.2)	694 (21.6)	630 (23.0)	
≥3	2164 (36.4)	1088 (33.9)	1076 (39.2)	
Missing	1284	669	615	
ADLs				<0.001 ^a^
No	5630 (78.5)	3152 (82.0)	2478 (74.4)	
Yes	1544 (21.5)	693 (18.0)	851 (25.6)	
Missing	62	34	28	
Living Near Children				0.001 ^a^
No	3313 (45.8)	1849 (47.7)	1464 (43.6)	
Yes	3923 (54.2)	2030 (52.3)	1893 (56.4)	
Economic Support				0.010 ^a^
No	504 (7.0)	298 (7.7)	206 (43.6)	
Yes	6732 (93.0)	3581 (92.3)	3151 (93.9)	
Weekly Contact				<0.001 ^a^
No	1570 (21.7)	930 (24.0)	640 (19.1)	
Yes	5663 (78.3)	2946 (76.0)	2717 (80.9)	
Missing	3	3	0	
Cognitive function				<0.001 ^c^
Mean (SD)	13.97 (5.50)	14.63 (5.09)	13.20 (5.84)	

N.B. Total percentages of some variables are not equal to 100 due to rounding. ^a^—Outcomes of χ^2^, ^b^—outcomes of Wilcoxon rank sum test, ^c^—outcomes of independent sample *t*-test.

**Table 4 ijerph-18-00021-t004:** Patterns of grandparent caregiving in male vs. female respondents.

Variable	Total	Men	Women	*p*
	(*n* = 7236)	(*n* = 3879)	(*n* = 3357)	
	*n* (%)	*n* (%)	*n* (%)	
Care Grandchildren				0.477 ^a^
No	3408 (47.1)	1842 (47.5)	1566 (46.6)	
Yes	3828 (52.9)	2037 (52.5)	1791 (53.4)	
Frequency				<0.001 ^a^
Never	3408 (50.9)	1842 (52.7)	1566 (49.0)	
Not regularly	1753 (26.2)	910 (26.0)	843 (26.4)	
Regularly	1533 (22.9)	745 (21.3)	788 (24.7)	
Missing	294	151	143	
Intensity				<0.001 ^a^
Never	3408 (51.0)	1842 (52.8)	1566 (49.0)	
Moderate grandchild care	1300 (19.5)	723 (20.7)	577 (18.0)	
Intensive grandchild care	1973 (29.5)	923 (26.5)	1050 (32.9)	
Missing	274	140	134	
Number of Grandchildren				0.135 ^a^
None	3408 (47.7)	1842 (48.1)	1566 (47.2)	
1	3004 (42.0)	1576 (41.1)	1428 (43.1)	
≥2	735 (10.3)	414 (10.8)	321 (9.7)	

N.B. Total percentages of some variables are not equal to 100 due to rounding. ^a^—Outcomes of χ^2^.

**Table 5 ijerph-18-00021-t005:** Cognitive function in different patterns of grandparent caregiving.

Variable	Total (*n* = 7236)	*p*	Men (*n* = 3879)	*p*	Women (*n* = 3357)	*p*
	Mean (SD)		Mean (SD)		Mean (SD)	
Care Grandchildren						
No	13.01 (5.66)	<0.001 ^c^	13.84 (5.24)	<0.001 ^c^	12.04 (5.96)	<0.001 ^c^
Yes	14.82 (5.21)		15.35 (4.85)		14.22 (5.54)	
Frequency						
Never	13.16 (5.64)	<0.001 ^d^	13.98 (5.23)	<0.001 ^d^	12.07 (5.98)	<0.001 ^d^
Not regularly	15.12 (5.02)		15.58 (4.67)		14.56 (5.35)	
Regularly	14.81 (5.23)		15.45 (4.78)		14.18 (5.56)	
Intensity						
Never	13.16 (5.64)	<0.001 ^d^	13.99 (5.22)	<0.001 ^d^	12.09 (5.97)	<0.001 ^d^
Moderate grandchild care	14.69 (5.27)		15.44 (4.72)		13.74 (5.76)	
Intensive grandchild care	14.98 (5.12)		15.50 (4.81)		14.52 (5.35)	
Number of Grandchildren						
None	13.05 (5.65)	<0.001 ^d^	13.87 (5.23)	<0.001 ^d^	12.10 (5.97)	<0.001 ^d^
1	14.99 (5.14)		15.52 (4.76)		14.41 (5.48)	
≥2	14.13 (5.42)		14.74 (5.16)		13.35 (5.65)	

N.B. Total percentages of some variables are not equal to 100 due to rounding. ^c^—Outcomes of independent sample *t*-test, ^d^—outcomes of one-way ANOVA test.

**Table 6 ijerph-18-00021-t006:** Associations between cognition and patterns of grandparent caregiving.

Variable	Model 1: Total	Model 2: Men	Model 3: Women
	β (SE)	β (SE)	β (SE)
	(95% CI)	(95% CI)	(95% CI)
Care Grandchildren (reference: No)	0.692 *** (0.179)	0.703 ** (0.233)	0.744 ** (0.275)
	(0.340, 1.043)	(0.245, 1.160)	(0.203, 1.284)
Care Grandchildren × Age	0.068 ** (0.021)	0.099 *** (0.029)	0.045 (0.032)
	(0.026, 0.110)	(0.043, 0.155)	(−0.017, 0.108)
Age	−0.140 *** (0.015)	−0.154 *** (0.020)	−0.136 *** (0.023)
	(−0.170, −0.110)	(−0.194, −0.114)	(−0.181, −0.090)
Intensity (reference: Never)			
Moderate grandchild care	0.965 *** (0.239)	0.797 ** (0.309)	1.346 *** (0.371)
	(0.496, 1.434)	(0.191, 1.403)	(0.618, 2.074)
Moderate grandchild care × Age	0.102 *** (0.031)	0.138 *** (0.039)	0.066 (0.049)
	(0.042, 0.162)	(0.061, 0.215)	(−0.030, 0.162)
Intensive grandchild care	0.456 * (0.214)	0.619 * (0.286)	0.334 (0.318)
	(0.037, 0.875)	(0.057, 1.181)	(−0.290, 0.958)
Intensive grandchild care × Age	0.040 (0.027)	0.058 (0.038)	0.027 (0.038)
	(−0.012, 0.092)	(−0.016, 0.132)	(−0.047, 0.102)
Age	−0.142 *** (0.015)	−0.154 *** (0.020)	−0.139 *** (0.023)
	(−0.171, −0.113)	(−0.193, −0.116)	(−0.184, −0.094)
Frequency (reference: Never)			
Not Regularly	0.807 *** (0.222)	0.633 * (0.292)	1.051 ** (0.334)
	(0.372, 1.241)	(0.061, 1.205)	(0.395, 1.707)
Not Regularly × Age	0.049 ^+^ (0.027)	0.096 * (0.038)	0.025 (0.041)
	(−0.005, 0.103)	(0.022, 0.170)	(−0.054, 0.105)
Regularly	0.578 * (0.227)	0.880 ** (0.300)	0.376 (0.344)
	(0.132, 1.023)	(0.292, 1.468)	(−0.299, 1.050)
Regularly × Age	0.093 ** (0.029)	0.111 **(0.039)	0.067 (0.044)
	(0.036, 0.150)	(0.034, 0.188)	(−0.018, 0.153)
Age	−0.143 *** (0.015)	−0.158 *** (0.020)	−0.139 *** (0.023)
	(−0.173, −0.113)	(−0.197, −0.119)	(−0.184, −0.094)
Number of Grandchildren (reference: No)			
1 child	0.677 *** (0.190)	0.620 * (0.247)	0.788 ** (0.291)
	(0.305, 1.049)	(0.135, 1.105)	(0.218, 1.359)
1 child × Age	0.058 * (0.022)	0.089 ** (0.031)	0.041 (0.035)
	(0.013, 0.103)	(0.028, 0.149)	(−0.026, 0.109)
≥2 children	0.556 ^+^ (0.302)	0.755 ^+^ (0.403)	0.626 (0.461)
	(−0.035, 1.148)	(−0.035, 1.545)	(−0.278, 1.529)
≥2 children × Age	0.068 (0.042)	0.115 * (0.058)	0.019 (0.060)
	(−0.013, 0.150)	(0.002, 0.229)	(−0.097, 0.136)
Age	−0.130 *** (0.015)	−0.150 *** (0.020)	−0.132 *** (0.023)
	(−0.165, −0.106)	(−0.189, −0.111)	(−0.177, −0.087)
R^2^_adjusted_	0.33~0.35	0.26~0.28	0.40~0.41
VIF	1~3	1~3	1~3

N.B. All models were adjusted for age, education, marital status, residence, retirement, personal income, drinking, smoking, number of types of chronic diseases, ADLs, living near children, economic support, and weekly contact; ^+^: *p* < 0.1; *: *p* < 0.05; **: *p* < 0.01; ***: *p* < 0.001.

## Data Availability

Datasets are distributable only by CHARLS team. They are available in the public domain through 395 registrations on CHARLS website: http://charls.pku.edu.cn/zh-CN and are also available on request from the corresponding author.

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
