# Peer review of "Intergenerational Ties in Context: Association between Caring for Grandchildren and Cognitive Function in Middle-Aged and Older Chinese"

_ijerph, 2020, doi:10.3390/ijerph18010021_

Round 1
Reviewer 1 Report
I am satisfied with the authors' revisions and appreciate their efforts.
Reviewer 2 Report
Authors have significantly improved the quality of the paper.
Reviewer 3 Report
As far as I can see, all of my previous comments have been duly considered and appropriate adjustments of the manuscript done
This manuscript is a resubmission of an earlier submission. The following is a list of the peer review reports and author responses from that submission.
Round 1
Reviewer 1 Report
The current manuscript examines the impact of caregiving by grandparents on their cognitive performance. The unique population, inclusion of gender, and separation of caregiving variables which are especially impactful for custodial grandparents are significant strengths of the reviewed manuscript. However, there are currently a few limitations that, if addressed, I believe will strengthen the manuscript and increase its potential impact.
- The first limitation of the manuscript is its readability. The authors do not appear to be native English speakers, which is reflected in the transitions, grammar, and overall readability of the text. I recommend that they either bring on a native English speaker as a coauthor and have that person edit the manuscript or at least use a thorough grammatical editing service like Grammarly Premium before resubmission.
- Another significant issue of concern is the project’s clinical significance. Looking at Table 4, it appears that the effect sizes (which are not reported within the text) are rather small. With a sample size of 7,236 participants, finding significant group differences comes easily. It is important to report the effect size and justify whether these differences are clinically meaningful. The potential impact of the manuscript would be greatly enhanced by providing a measure of effect size, as well as a table with the means and SDs for cognitive performance broken down by caregiver type, and the percentage of each caregiver type that fell below established cut-offs.
- Speaking of established cut-offs, it would be helpful to have much more detail provided regarding the assessment and scoring technique used for the Mini-Mental State Examination scores. The MMSE is designed to have a range of 0 – 30, with 7 domains, and to employ the cut-offs of a score from 24 + indicating no impairment, 18 – 23 indicating mild impairment, and below 18 indicating severe impairment. However, the authors report a possible range of up to 31 points for the MMSE (where did the extra point come from?), only four cognitive domains (only one of which directly matches the domains outlined within the scoring of the MMSE), and although they do not report the percentage of the sample that falls below the cut-off points, they report means by gender ranging from 13.20 to 14.63, both of which would fall into the severe cognitive impairment range of the MMSE. Please clarify the scoring system used within the manuscript, and justify differences made from the scoring criterion outlined for the use of the MMSE by the Psychological Assessment Resources group, as well as outlining how the performance of your sample compares to the measure cut-offs for cognitive impairment. Perhaps these differences are unique to the Chinese version of the MMSE, but if so, this needs to be explained more fully within the document.
- Please provide a justification as to the “intensity” and “frequency” categories of grandchild care used within the study. The authors mention that the “low” intensity grouping of anything less than 40 hours per week was based around a standard workweek. However, most of these grandparents are not earning income for the care of their grandchildren (which means that they may be employed elsewhere), and other studies have shown an average of 12 hours per week spent by grandparents on childcare duties (Council of the Ageing Report, 2012). 39 hours per week is quite high. Why not use a quartile approach from a frequency distribution unique to your sample? The same can be said for the “frequency” variable. The “low” frequency group cares for their grandchildren up to half of each year! Does this match the average annual caregiving time spent by grandparents reported elsewhere?
- Please be sure to spell out all abbreviations before using them within the text (e.g., NCD).
- What were the proposed hypotheses? Page 2 mentions the study’s aim, but no predictions/hypotheses were reported. Please list all apriori hypotheses and report on their support from your findings within the discussion section.
Thank you for the opportunity to review this interesting and meaningful work. I look forward to its eventual publication.
Author Response
Dear Reviewer,
Title: Intergenerational Ties in Context: Association between Caring for Grandchildren and Cognitive Function in Middle- and Old-Aged Chinese
Thank you very much for the reviewers’ reports. We deeply appreciate the valuable comments you have provided. We agree that we should have provided a more comprehensive and insightful view of the relationship between caring for grandchildren and cognitive function among middle- and older-aged Chinese. We have revised the original manuscript accordingly and carefully proof-read the manuscript to minimize typographical, grammatical, and bibliographical errors. Here, we attached the revised manuscript in the format of MS word for your review, and we have highlighted the sections with major revision and minor corrections by using the “Track Changes” function in Microsoft Word. In this document, we have highlighted the sections with major revisions in italics.
Below is our response to their comments resulting in a number of clarifications. Revisions have been highlighted in the revised manuscript.

Reviewer 2 Report
typing miscue, vg. “chinses” instead of chineses; “(MMSE) was to measure” instead of “(MMSE) was used to measure”. Characteristics of the sample is not described in the abstract.
The statistical analysis does not seem appropriate fo the analysis. It reads “multivariate linear regression” but based on the hypothesis should be an ANOVA were cognition performance would fall as dependent variable.
I fear, MMSE is not a suitable scale to explore a wide range of variability in cognitive function since it is aim to detect cognitive impairing for example in screening for dementia.
The age range is too broad. It does not seem reasonable to expect that grandparenting will have the same effects on 45 and 100 years old person.
In sum, I do not feel the conclusions are supported by the data.
Author Response

(The authors gave the same response as above.)

Reviewer 3 Report
My general impression is that this is a well conducted study that could be published in the IJERPH, but whcih wold profit from some moderate changes.
Mostly in the introduction and discussion, a thorough review of syntax and grammar and splitting of some long sentences would make the text more reader-friendly.
In the abstract, omission of the parentheses with numbers and just pointing to trends and findings that were positive or negative and significant or not, without showing the detailed figures, would make it more readable.
Lines:
49: The expression 'intensive care' is commonly used for what is done in hospital intensive care units. I assume the meaning here is different and a term that better describes what is meant, should be considered.
77: ..(2015).. should probably be a reference of publication (45)?
94: ..Zhao et al... The standard is to refer to the first author, which is Lei. So this should be ...Lei et al....
136-142:..a greater proportion....... The wording here is as if a single proportion had all the characteristics mentioned. ..Single proportions were.... would be more correct.
Tables 2 and 3: For about half of the characteristics listed , the numbers listed do not add up to the total of participants given in the column heading. The groups of non-responders / data not available should at least be acknowledged in the text, possibly included in the table. The percentages should be given with a single decimal only.
163-177: The text describing the findings presented in the tables shold be verbal comments to the findings, pointing out trends and direction of the findings, significant and non-significant trends and differences, but not a repetition of the exact findings shown in the table. Thus, the parentheses repeating the numbers of the table should be omitted, and the verbal comments a little more extensive, pointing to findings that are subjects for discussion under that heading. For example, the finding that non-intensive care care is more strongly associated to cognition than intensive care should be pointed out (and a finding that should be discussed in the discussion). This would make the presentation of findings clearer and easier to read.
Section 4.2. The relationship..........: Throughout the section cognition is discussed as the response variable and all other variables as explanatory variables. It would seem to be appropriate also to give some consideration to the possibility of opposite direction of causality, with the impact of mental capacity on some of the exlanatory variables.
line 244: I doubt the validity of this statement on the basis of the results of the present study.
248-9: '..rather than...themselves'. As this is not a choice available to post-menopausal women, the statement should be omitted.
312:..that the association.. change to: ..that a statistically significant association...
Author Response

(The authors gave the same response as above.)
